# A Momentum Contrastive Learning Framework for Low-Data Wafer Defect Classification in Semiconductor Manufacturing

Yi Wang [1], Dong Ni [1,*] and Zhenyu Huang [2]

1   College of Control Science and Engineering, Zhejiang University, Hangzhou 310027, China; 11732019@zju.edu.cn
2   Intel Corporation, Dalian 116630, China; zhenyu.huang@intel.com
*   Correspondence: dni@zju.edu.cn

**Abstract:** Wafer bin maps (WBMs) are essential test data in semiconductor manufacturing. WBM defect classification can provide critical information for the improvement of manufacturing processes and yield. Although deep-learning-based automatic defect classification models have demonstrated promising results in recent years, they require a substantial amount of labeled data for training, and manual labeling is time-consuming. Such limitations impede the practical application of existing algorithms. This study introduces a low-data defect classification algorithm based on contrastive learning. By employing momentum contrastive learning, the network extracts effective representations from large-scale unlabeled WBMs. Subsequently, a prototypical network is utilized for fine-tuning with only a minimal amount of labeled data to achieve low-data classification. Experimental results reveal that the momentum contrastive learning method improves the defect classification performance by learning feature representation from large-scale unlabeled data. The proposed method attains satisfactory classification accuracy using a limited amount of labeled data and surpasses other comparative methods in performance. This approach allows for the exploitation of information derived from large-scale unlabeled data, significantly reducing the reliance on labeled data.

**Keywords:** contrastive learning; low data; self-supervised learning; wafer bin map; defect classification; semiconductor manufacturing





## 1. Introduction

With the advancement of the integrated circuit industry, semiconductor manufacturing processes have become increasingly complex. Chip production involves hundreds of types of processing equipment and numerous processing steps. Variations in the manufacturing process and environment can lead to defects, resulting in significant uncertainties in production yield, directly impacting a company's costs and profits. Consequently, it is imperative to reduce process defects, decrease losses in semiconductor manufacturing, and enhance production efficiency and economic benefits.

A wafer is a fundamental unit in integrated circuit manufacturing. After the manufacturing process, testing equipment evaluates the functionality of each die (chip) on the wafer. Hundreds of dies on a wafer are tested using chip probing. Furthermore, the die with a passing testing result will be packaged as a final chip. Figure 1 presents the relationship between wafer, die, and chip.

Wafer bin maps (WBMs) represent the test results of the wafer based on the pass or fail (bin) values of each die. Multiple defective dies form spatial patterns as defect patterns. We refer to such instances where multiple defective dies are spatially aggregated, forming spatial patterns as wafer defects. Each defect pattern is associated with a failure or anomaly in a specific segment of the production line. Thus, WBM defect classification can provide critical information for the improvement of manufacturing processes and yield [1]. Figure 2 presents examples of several wafer bin maps with specific defect patterns. Black dots represent failed dies and white dots represent passing dies.

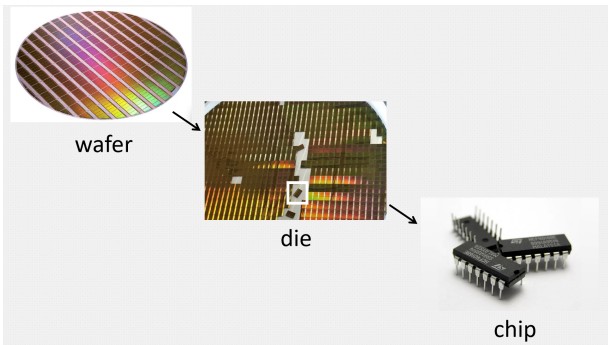

**Figure 1.** Relationship between wafer, die, and chip.

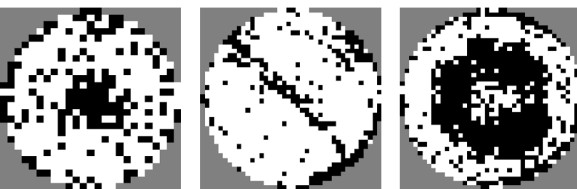

**Figure 2.** Examples of several WBMs with specific defect patterns.

In most semiconductor manufacturing enterprises, WBM defect classification primarily relies on human expertise. Manual labeling is time-consuming and can be subject to variations due to the subjective judgments of different engineers [2]. Therefore, investigating methods for automatic defect classification is of paramount importance. In this regard, researchers have proposed numerous automatic defect classification approaches over the past two decades, including statistic learning-based methods [3–5] and machine-learning-based methods [6–12].

In recent years, defect classification algorithms based on deep learning have been gradually proposed and developed because they improved image classification performance. T. Nakazawa [13] employed convolutional neural networks (CNNs) for WBM defect classification. S. Cheon [14] developed an automatic defect classification (ADC) method based on deep learning, capable of automatically classifying different wafer surface damage types. The proposed method could accurately identify the GFA classes that were not encountered during model training by comparing the CNN features of the unseen classes. M. Saqlain [15] proposed an ensemble convolutional neural network (ECNN) framework for WBM defect classification, adopting a weighted majority function to assign higher weights to the base classifiers with superior predictive performance. Furthermore, K. Kyeong [16] suggested using CNNs to classify mixed-type GFA in WBMs in an individual classification model framework for each GFA.

However, training deep learning classification methods requires a large amount of labeled data, which meets the conflict that manual labeling is time-consuming and inconsistent. Moreover, in practice, there exists a vast amount of unlabeled WBMs that have not been utilized. Therefore, making good use of the large number of unlabeled WBMs and reducing the reliance on labeled data is one of the pressing issues that need to be addressed. Several methods have been proposed for utilizing unlabeled WBMs, such as semisupervised learning. Kong and Ni [17] proposed to adopt a ladder network and the semisupervised variational autoencoder to classify WBMs. Active learning and pseudo-labeling are also utilized to accelerate learning.

Recently, self-supervised learning has been proposed as an effective unsupervised learning branch. Self-supervised learning is a type of machine learning where a model learns to represent data in a way that is useful for downstream tasks without requiring direct supervision or labeled data [18]. Instead, the model is trained on an auxiliary task designed to capture meaningful patterns in the data. This approach is advantageous when labeled data are scarce or expensive, as it allows the model to learn from large amounts of

unlabeled data. Based on the pretext task for visual features in images, self-supervised learning approaches can be classified into three categories: generation-based [19–23], context-based [24–29], and contrastive-based [30–36].

In WBM yield, H. Kahng and S.B. Kim [2] proposed a self-supervised learning-based framework that uses unlabeled data to learn rich visual representations beforehand to realize a data-efficient WBM GFA classification. D. Kim and P. Kang [37] proposed a dynamic WBM clustering method using pseudo-labels. H. Geng [38] proposed an end-to-end wafer defect classifier that unites the few-shot learning and self-supervised learning algorithms in the training period, which opens up a new line.

The existing research on WBM self-supervised learning effectively leverages unlabeled data and enhances classification performance. However, it still requires a substantial amount of labeled data for fine-tuning. Improving the feature representation ability of the model is the critical way to reduce the demand for labeled data. A more effective self-supervised learning method is needed to utilize the information in unlabeled data fully, enhance the model's feature representation ability, and improve its classification performance with limited labeled data. To this end, this study introduced momentum contrastive learning [32] for self-supervised pretraining on unlabeled WBMs. Contrastive learning is a branch of self-supervised learning that learns representations by maximizing the agreement between different views of the same data point and has been shown to achieve state-of-the-art performance in many computer vision tasks, especially in low-data regimes. Momentum contrastive learning is a variant of contrastive learning in which a momentum encoder is introduced to improve the quality of learned representations. The momentum encoder is updated using an exponential moving average of the parameters from the primary encoder, which encourages the momentum encoder to capture the global structure of the data while the primary encoder focuses on capturing local features. By leveraging the momentum encoder, momentum contrastive learning performs better than standard contrastive learning in various computer vision tasks.

After momentum contrastive learning, we proposed a low-data fine-tuning method for WBM defect classification. Thanks to H. Geng [38], we reference it to use a prototypical few-shot learning approach. Moreover, different to [38], the prototypical network is only performed in the fine-tuning period in the proposed framework. By comparing with other existing methods, the proposed method outperforms the comparative methods regarding classification performance with real-world WBM data. The results indicate that introducing momentum contrastive learning improves the performance of supervised learning and makes it possible to achieve classification with only a small amount of labeled data.

The contributions of the article are as follows:

1. Momentum contrastive learning is introduced for the unlabeled WBMs pretraining, enhancing the feature representation ability and improving WBM defect classification accuracy.
2. We propose a two-step method where the prototype network is fine-tuned after contrastive learning pretraining, improving low-data fine-tuning performance.
3. We utilize a real-world dataset and compare our proposed method with other existing WBM self-supervised learning algorithms, demonstrating superior defect classification performance.

## 2. Methods

In this study, we design a self-supervised pretraining framework based on momentum contrastive learning for large-scale unlabeled WBMs. After pretraining, wafer defect classification is achieved by fine-tuning with labeled data. In order to improve the efficiency of labeled data fine-tuning, a few-shot learning method is adopted during fine-tuning. The overall flowchart is shown in Figure 3. In the following text, we introduce each block and the overall framework in detail.

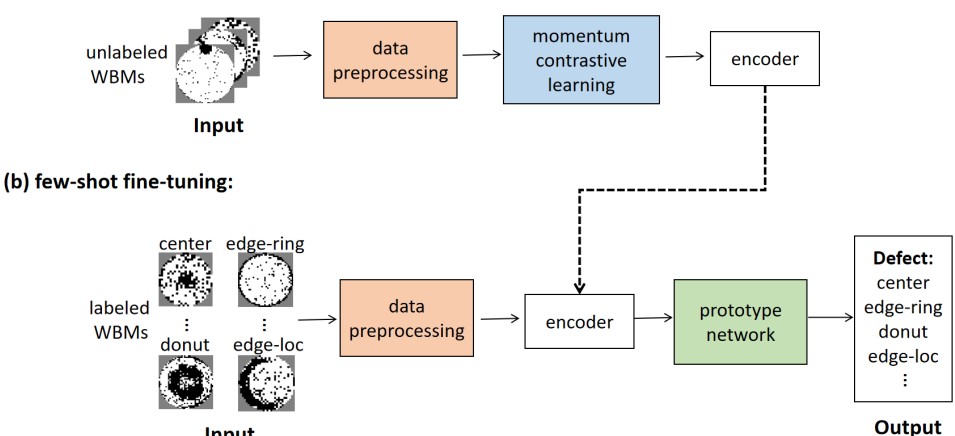

**Figure 3.** The proposed two-stage flowchart.

*2.1. Data Preprocessing*

In this study, the data preprocessing period mainly comprises resizing and denoising. The real-world wafer bin map dataset contains WBMs of slightly different sizes. To ensure consistent input dimensions for subsequent modeling, the original WBMs are resized using nearest-neighbor interpolation. In this study, all wafer maps are resized to $64 \times 64$.

The WBM defect pattern can be seen as a superposition of two independent components: random noises (particle correlation) and system clustering (process correlation). Random noises are generally problems with the clean room environment and tend to rise and fall with the overall cleanliness of the clean room. Reducing random noises requires long-term, incremental improvements or expensive equipment overhauls, but systematic defects can be easily eliminated through corrective efforts. Conversely, system defects caused by assignable causes are often attributed to processing equipment and human error. Therefore, it is necessary to remove random noise dies before conducting defect pattern analysis.

To reduce random dies, we applied the median filtering algorithm adapted to WBM denoising. The median filtering algorithm effectively removes various types of noise, including salt-and-pepper and Gaussian noise. It is also able to preserve image edges and details better than other types of smoothing filters. The algorithm works by smoothing the image using the median value of the pixels in a sliding window. Specifically, a window of size $N \times N$ is centered around each pixel in the image. The pixel values in the window are sorted in ascending order, and the median value is then assigned to the central pixel. In this study, we set $N$ equal to three.

*2.2. Momentum Contrastive Learning*

For large-scale unlabeled WBMs, the momentum contrastive learning method is introduced in this study for self-supervised learning. Momentum contrastive learning is designed to improve the training efficiency of contrastive learning algorithms by introducing a momentum-based update to the model weights, and it is effective in various applications, such as image classification, object detection, and natural language processing. It is advantageous when labeled data are limited or unavailable, as it generates high-quality feature representations without manual annotation.

The proposed momentum contrastive learning framework is presented in Figure 4. Contrastive learning [32] can be regarded as training an encoder for a dictionary look-up task. It generates two augmented views for each WBM in a batch randomly sampled from a WBM dataset. These two views are then projected into an embedding space, where a self-supervised loss is applied to train the encoder and generate effective representations.

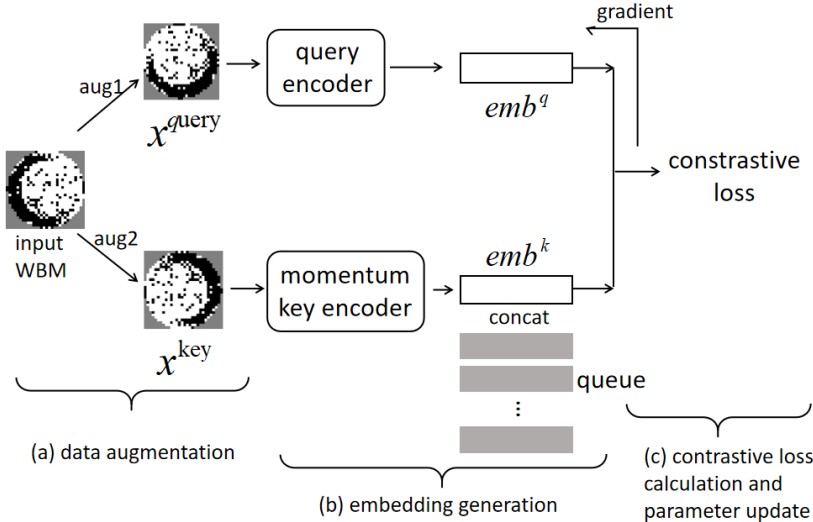

**Figure 4.** The proposed momentum contrastive learning framework.

Momentum contrastive learning consists of three parts. The first part is the data augmentation module, which generates two correlated views of each WBM. To facilitate WBM defect classification, we apply simple augmentations such as rotation and flipping. These augmentations are designed to preserve the underlying failure patterns and do not alter the latent categories of the data.

The second part is the embedding generation part. The input of this part is the query (one augmented WBM) and the key (another augmented WBM) of each WBM. Two encoders transform the WBMs from two augmented views to embedding. As ResNet has been shown to achieve state-of-the-art results on various computer vision tasks, we applied ResNet-18 [39] as the backbone of the two encoders.

Moreover, the queue is a significant development of momentum contrastive learning, shown in the second part. Contrastive learning involves constructing a discrete dictionary from high-dimensional inputs representing a sampled subset of all data. In momentum contrastive learning, the dictionary is maintained as a queue of data embedding. Using a queue, it decouples the dictionary size $K$ from the batch size, which can be much larger than the batch size. Additionally, the queue is renewed after each epoch. At the end of each training episode, the current batch is enqueued to the dictionary, and the oldest batch in the queue is removed to maintain a fixed queue size.

The third part is contrastive loss calculation and parameter update. The queries and their corresponding keys are encoded for the current batch, forming positive sample pairs. The negative samples are all the samples in the queue. A contrastive loss is a function whose value is low when the query is similar to its corresponding positive key and dissimilar to all other keys in the queue. The contrastive loss function is InfoNCE [32]:

$$L_q = -\log \frac{\exp(q \cdot k_+ / \tau)}{\sum_{i=0}^{K} \exp(q \cdot k_i / \tau)}, \tag{1}$$

where $q$ is the query and $k_+$ is the positive key of $q$. $k_i$ represents the keys in the queue.

The second development of momentum contrastive learning is the momentum update. In the backpropagation period, the gradient only propagates to the queries, and only the parameters of the query encoder are updated by backpropagation. Then, the parameters of the key encoder are updated by:

$$\theta_k \leftarrow m\theta_k + (1 - m)\theta_q \tag{2}$$

Here, $m$ is a momentum coefficient. $\theta_k$ and $\theta_q$ represent the parameters of the momentum encoder and query encoder, respectively. We default to $m = 0.999$ because a relatively

large momentum works much better than a smaller value. It is suggested that a slowly evolving key encoder is core to using a queue [32].

### 2.3. Few-Shot Fine-Tuning

After self-supervised pretraining, in order to perform the classification task, it is necessary to fine-tune the query encoder on a large number of labeled WBMs. However, manually labeling WBMs is highly costly and time-consuming. This study proposes a few-shot fine-tuning method that uses a prototypical network in the fine-tuning period, which can significantly improve the efficiency of labeled data and reduce the demand for labeled samples.

The prototypical network is an effective few-shot classification method with high accuracy with limited labeled samples. In the prototypical network, a small-scale labeled WBM dataset $L$ is given as $L = \{(x_1, y_1), \cdots, (x_N, y_N)\}$, where each $x_i \in R^L$ is the $D$—dimensional embedding of a WBM and $y_i \in \{1, \cdots, K\}$ is the label. $L_k$ denotes the WBMs set labeled with class $k$.

Then, a prototype is an $M$—dimensional representation $c^k \in R^M$ of each class through the query encoder function $f_\phi : R^D \rightarrow R^M$ with learnable parameters $\phi$. Each prototype is the mean of the embedded support set belonging to its class [40]:

$$c_k = \frac{1}{L_k} \sum_{(x_i, y_i) \in L_k} f_\phi(x_i) \tag{3}$$

Prototypical networks generate a probability distribution over classes for a given query point $x$ by applying the softmax function on the distances between the query point and the class prototypes in the embedding space. The loss function is the negative log-probability:

$$J(\phi) = -\log \frac{\exp(-d(f_\phi(x), c_k))}{\sum_{k'} \exp(-d(f_\phi(x), c_{k'}))} \tag{4}$$

The squared Euclidean distance is adopted as the distance metric:

$$-d(f_\phi(x), c_k) = \| f_\phi(x) - c_k \|_2^2 . \tag{5}$$

Fine-tuning the query encoder is achieved by minimizing the loss function of the true class $k$ via stochastic gradient descent (SGD) [40]. In the training process, a subset of classes from the training dataset is randomly selected in each episode. For each selected class, a subset of WBMs is chosen to form the support set, while a subset of the remaining WBMs is selected as query points. The prototypical network procedure is shown in Figure 5. We set the sample number of a support set as five and the class number as three. After fine-tuning, the classification was performed by computing the distance between the input and the prototypes.

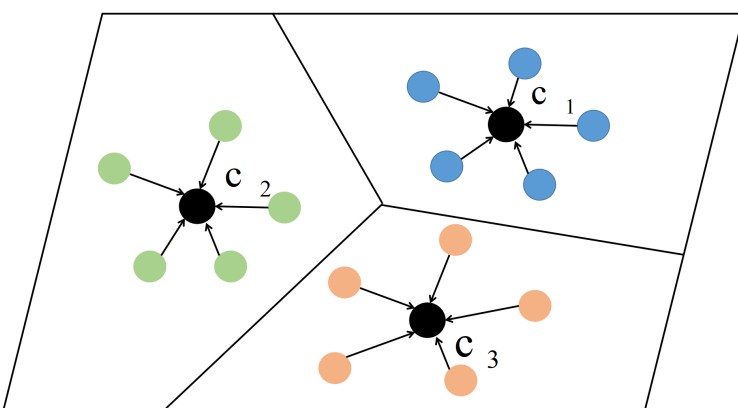

**Figure 5.** The prototypical-network-based few-shot procedure.

*2.4. Overall Framework*

As illustrated in Figure 3, the framework can be divided into two stages. The first stage is the self-supervised pretraining stage, which inputs unlabeled WBMs. First, data preprocessing is performed in this stage, including WBM denoising and resizing. Then, momentum contrast learning is used to train the two encoders using queries and keys generated from two augmented views of the input WBMs. The query and key encoder extract feature representations of queries and keys and output encoded embedding. A WBM dictionary called "queue" is introduced in momentum contrastive learning to reuse the encoded keys from the previous batches. Finally, the query and the corresponding key embedding are positive pairs. The query embedding and the queue are negative pairs for calculating the contrastive loss. Only the parameters of the query encoder are updated by backpropagation. The parameters of the key encoder are updated by the momentum mechanism.

The second stage is the fine-tuning stage, which employs labeled WBMs to fine-tune the pretrained encoder for classification. Similar to the pretraining stage, data preprocessing is carried out initially. During the fine-tuning process, a prototypical network is utilized for classification. Ultimately, a well-trained defect classification model is obtained.

## 3. Experiments and Discussion

This section introduces the experiments of evaluating the performance of the proposed framework, including the experiment data, performance evaluation of the overall framework, and ablation study of momentum contrastive learning.

*3.1. Data and Setup*

In this study, we employed a real-world WBM dataset, WM-811K, for experiments, which is publicly available. WM-811K included 811,457 wafer maps from 46,293 lots, covering nine categories of patterns, and only 172,950 were annotated by experts. The other 638,507 wafer maps are unlabeled WBMs. In our self-supervised pretraining period, 500,000 unlabeled WBMs were used for momentum contrastive learning. The labeled dataset consisted of nine WBM defect types, including eight systematic patterns and one with no systematic pattern (none). The examples of nine existing defect types in WM-811K are shown in Figure 6.

The WM-811K consists of binary WBMs with only pass (0) or fails dies (1). Each wafer map was treated as a grayscale two-dimensional image. For evaluating the defect classification performance of the proposed method, we compared the proposed method with two existing WBM classification methods. To ensure a fair comparison, the labeled dataset must remain the same in the comparison methods and was split into 0.6:0.4 for training and test set. Table 1 presents the statistics of the labeled dataset.

**Table 1.** WM-811K data description.

| Defect Type | All Numbers | Train (0.6) | Test (0.4) | Percent (%) |
|---|---|---|---|---|
| Center | 4294 | 2576 | 1718 | 2.48 |
| Donut | 555 | 333 | 222 | 0.32 |
| Edge-loc | 5189 | 3113 | 2076 | 2.99 |
| Edge-ring | 9680 | 5808 | 3872 | 5.60 |
| Location | 3593 | 2156 | 1437 | 2.08 |
| Near-full | 149 | 89 | 60 | 0.09 |
| Random | 866 | 520 | 346 | 0.50 |
| Scratch | 1193 | 716 | 477 | 0.69 |
| None | 147,431 | 88,459 | 58,972 | 85.25 |

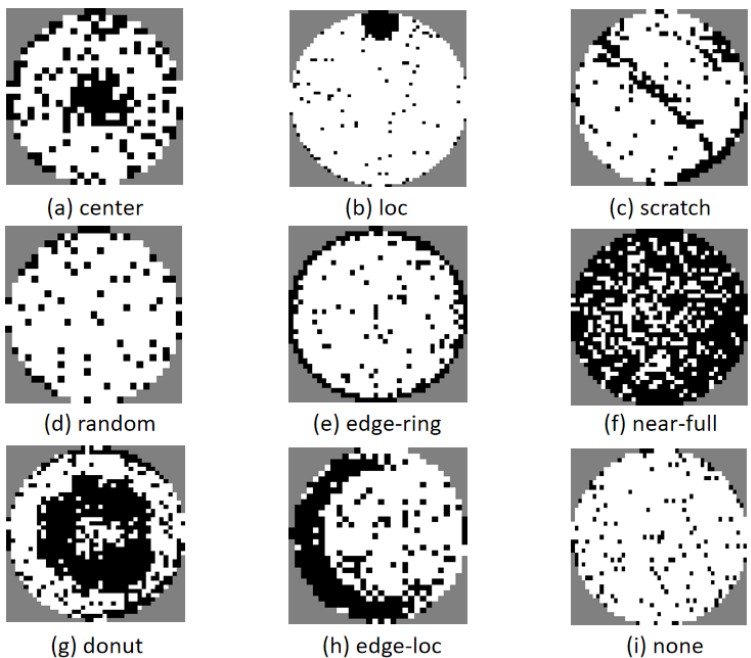

**Figure 6.** The examples of 9 defect types in the WM-811K dataset.

For momentum contrastive learning, we adopted ResNet-18 as the query and key encoder's backbone. In this work, the hyperparameters were chosen by manual tuning. We compared the performance of several common learning rate values: 0.1, 0.01, and 0.001. Moreover, we compared typical batch sizes from a smaller value to the maximum value supported in the memory capacity: 64, 128, and 256. We also compared the performances of several different epochs. When the training loss no longer converged, the epoch was used as the minimum value, and it was increased by 100 sequentially, which was 300, 400, and 500, respectively. Finally, we chose the hyperparameters performing better than other candidates. The learning rate, batch size, and epoch were 0.01, 256, and 500. The encoder's output was a 1000-dimensional embedding. The momentum coefficient $m$ was set to 0.999, as described in Section 2.2. The length of the queue was 4096, the maximum value supported in the memory capacity. We used SGD as our optimizer. The SGD weight decay was 0.0001. We trained our model on four Nvidia V100 GPUs with 16 GB memory. The data augmentation was rotatedfor random angles.

For the prototypical-network-based fine-tuning, the hyperparameters were determined by the same approach as contrastive learning. The learning rate, batch size, and epoch were 0.01, 64, and 200. We used SGD as the optimizer. The original WM-811K dataset had 147,431 training data, which was too large for practical purposes. Furthermore, over 85% of these WBMs had no defect pattern ("none" type), with only 15% containing specific defect patterns, totaling 15,308 WBMs. We propose to remove the "none" type to evaluate the performance of WBMs with specific defect patterns. We focused only on the eight defect categories for classification, significantly reducing the number of training samples. We chose the labeled data from the eight defect categories to create training and testing sets. Moreover, to ensure a fair comparison with existing methods, we also applied nine types of train and test sets to keep the train and test set the same as the comparative methods. The number of support set samples was five, and the number of prototypes were nine and eight, respectively.

### 3.2. Overall Framework Performance Evaluation

To evaluate the performance of the proposed framework, we used 500,000 unlabeled data in WM-811K for momentum contrastive learning. The labeled training set was used to fine-tune the query encoder for classification based on the prototypical network. After

fine-tuning, the labeled test set was used to test the classification performance, reflecting the overall framework's performance.

We compared the proposed framework with an existing WBM classification method based on self-supervised learning and few-shot learning [38] (refers to "ICCAD'21"). The proposed framework consists of a shared backbone and two branches for the few-shot and self-supervised learners. This method is an end-to-end framework with two jointly performed tasks sharing the same wafer pattern feature extractor. The loss function combined the few-shot and self-supervised learner loss, constituting a total loss function. Unlike IC-CAD'21, our proposed framework is a two-stage method using a self-supervised learner for model pretraining and a few-shot learner for fine-tuning. Another existing deep-learning-based wafer defect classification method proposed in [41] was also compared with our proposed framework (refers to "DAC'20"). In [41], deep selective learning was exploited with distinct coverage on the testing dataset for the defect pattern classification. For a fair comparison, the train and test sets of the three comparative methods were the same. We calculated precision, recall, and f1-score to evaluate classification performance. The calculation function of the three indexes is as follows:

$$Precision = \frac{TP}{TP + FP} \tag{6}$$

$$Recall = \frac{TP}{TP + FN} \tag{7}$$

$$f1\text{-}score = \frac{2 * Precision * Recall}{Precision + Recall} \tag{8}$$

where $TP$ is the number of wafer defects of one defect type being correctly classified, $FP$ is the number of wafer defects of other defect types being classified to one defect type, $TN$ is the number of wafer defects of other defect types being classified to other defect types, and $FN$ is the number of wafer defects of one defect type being classified to other defect types.

Table 2 shows the classification results of our proposed method and two comparative methods with precision, recall, and f1-score and their corresponding macroaveraged values (i.e., arithmetic means). The highest value for each category under each evaluation indicator is bolded. The results of our proposed method for eight defect types (without "none") and nine defect types (with "none") are both presented in Table 2. For nine defect types, our method macroaveragely outperformed ICCAD'21 with 12.3% and 5.4% improvement on precision and recall rate and a 9.2% rise on f1-score. Moreover, it outperformed DAC'20 with a macroaveraged precision, recall, and f1-score of 21.1%, 20.0%, and 22.4%, respectively. There are two differences between our method and ICCAD'21. ICCAD'21 proposed a shared backbone of self-supervised and few-shot learning and united the two learners in one period [38]. We proposed to perform self-supervised learning and few-shot learning in two periods; contrastive learning was applied in the pretraining period, and then few-shot learning was embedded into the fine-tuning period. The second difference is the contrastive learning method. We used the momentum contrastive learning for self-supervised learning. The results indicate that performing self-supervised and few-shot learning in two steps benefits the classification. Furthermore, momentum contrastive learning is a more effective self-supervised learning method for WBMs.

Moreover, our proposed method for fine-tuning with eight defect types outperformed fine-tuning with nine defect types in the area of eight defects macroaverage recall and f1-score. These results illustrate that removing the "none" type in the fine-tuning period benefits the classification of specific defects.

To visualize the classification performance of each defect type, we present the confusion matrix of nine and eight defect types of our proposed method in Figures 7 and 8. As can be seen in the confusion matrices, the diagonal cells of our algorithm have darker colors. Furthermore, fewer classes are seriously confused in our proposed method. We can observe that the "location" classification performance in our proposed method is relatively lower.

This is because the "location" has a similar geometric feature to "center" and "donut". Moreover, "scratch" and "location" are often confused because the two defect types often appear on one WBM, causing a mixed-type problem. By observing the real-world dataset, we found that mixed-type defects often occur on a single WBM, with only one label assigned to a WBM, leading to model confusion and a classification performance decrease.

**Table 2.** Classification performance comparison with two comparative methods.

| Defect Types | DAC'20 | | | ICCAD'21 | | | Ours | | | Ours Eight Defects | | |
|---|---|---|---|---|---|---|---|---|---|---|---|---|
| | Precision | Recall | F1-Score | Precision | Recall | F1-Score | Precision | Recall | F1-Score | Precision | Recall | F1-Score |
| Center | **0.949** | 0.942 | **0.945** | 0.736 | **0.950** | 0.830 | 0.838 | 0.879 | 0.858 | 0.830 | 0.899 | 0.863 |
| Donut | 0.798 | 0.748 | 0.772 | 0.806 | 0.842 | 0.824 | **0.958** | 0.708 | 0.814 | **0.942** | 0.795 | **0.862** |
| Edge-Loc | 0.739 | 0.690 | 0.714 | 0.647 | 0.802 | 0.716 | **0.872** | **0.893** | **0.882** | 0.870 | 0.883 | 0.876 |
| Edge-Ring | **0.992** | 0.950 | 0.970 | **0.992** | 0.921 | 0.955 | 0.978 | 0.980 | **0.979** | 0.964 | **0.986** | 0.975 |
| Location | 0.191 | 0.627 | 0.293 | 0.605 | 0.720 | 0.658 | 0.631 | **0.746** | 0.684 | 0.664 | 0.743 | **0.701** |
| Near-Full | 0.697 | 0.383 | 0.495 | 0.810 | 0.867 | 0.840 | 0.975 | **0.984** | **0.979** | **0.981** | 0.968 | 0.975 |
| Random | 0.608 | 0.553 | 0.579 | 0.816 | 0.652 | 0.724 | **0.938** | 0.910 | 0.924 | 0.913 | **0.940** | **0.926** |
| Scratch | 0.127 | 0.287 | 0.176 | 0.474 | 0.701 | 0.565 | 0.940 | **0.853** | **0.894** | **0.941** | 0.848 | 0.892 |
| None | 0.985 | 0.927 | 0.955 | **0.986** | **0.967** | **0.977** | 0.855 | 0.959 | 0.904 | - | - | - |
| Macro-avg | 0.676 | 0.679 | 0.656 | 0.764 | 0.825 | 0.788 | **0.887** | **0.879** | **0.880** | - | - | - |
| Eight defects Macro-avg | 0.638 | 0.645 | 0.618 | 0.736 | 0.807 | 0.764 | **0.891** | 0.869 | 0.880 | 0.888 | **0.883** | **0.896** |

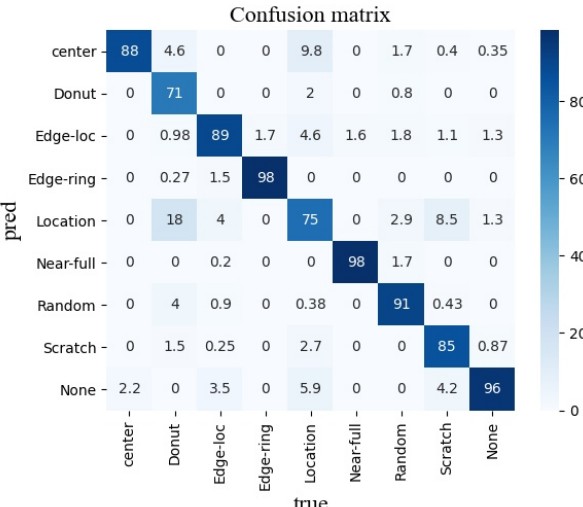

**Figure 7.** Confusion matrix of nine defect types in our proposed framework.

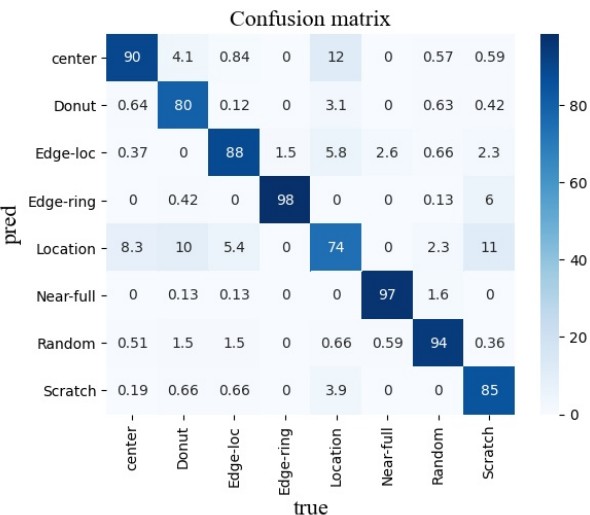

**Figure 8.** Confusion matrix of eight defect types in our proposed framework.

### 3.3. Ablation Study for Momentum Contrastive Learning

To independently evaluate the performance of momentum contrastive learning pretraining, we compared the classification performance of the query encoder with and without momentum contrastive learning pretraining. After pretraining, we attached two linear layers as a classification head to the end of the query encoder, and the classification results were obtained by softmax activation. Unlike the overall framework, we did not use the prototypical network as the classification mechanism during fine-tuning. For the case of using momentum contrastive learning pretraining, we pretrained the model using 500,000 unlabeled data with momentum contrastive learning. Then, we fine-tuned the classification model using the training set of the nine defect types labeled data, followed by testing the classification accuracy on the test set. For the case of not using momentum contrastive learning pretraining, we directly trained the classification model using the training set of the labeled dataset. Then, we tested the classification performance on the test set.

Table 3 shows the classification results with and without momentum contrastive learning pretraining. To verify the performance of momentum contrastive learning pretraining in the case of a small number of labeled WBMs, we also compared the classification performance when the number of labeled WBMs was reduced to 20%, 15%, and 10%, respectively. From Table 3, it can be seen that when using all labeled training sets, the recall of the classification model after momentum contrastive learning pretraining and fine-tuning was 85%. In contrast, the accuracy of the model trained directly without momentum contrastive learning pretraining was 80.7%, indicating that momentum contrastive learning pretraining improved the classification recall by 5%. When the number of labeled training samples was reduced to 20%, 15%, and 10%, the momentum contrastive learning increased the classification recall by 3%, 9%, and 3%, respectively, indicating that it significantly improved the performance in the case of a small number of labeled data.

**Table 3.** Classification recall comparison for with and without momentum contrastive pretraining.

| Percentage of Labeled Data | With Pretraining | Without Pretraining |
|:---:|:---:|:---:|
| 10% | 0.68 | 0.650 |
| 15% | 0.73 | 0.650 |
| 20% | 0.782 | 0.751 |
| All (100%) | 0.850 | 0.807 |

To visualize the improvement of using momentum contrastive learning, we plotted the results from Table 3 as a line chart, presented in Figure 9. This result also shows a significant improvement in classification performance when using momentum contrastive learning, which is observed even in the case of a reduced number of labeled WBMs. Moreover, it can be seen in Figure 9 that the classification performance of fine-tuning 10% data after momentum contrastive learning is about the same as training with 20% data without momentum contrastive learning. This indicates that using momentum contrastive learning significantly reduces the requirement of manual labeling and offers immense practical value for practice applications.

In order to accomplish higher accuracy, it is better to perform automatic tuning algorithms such as grid search and Bayesian optimization for hyperparameters. However, because momentum contrastive learning is computationally intensive and our computing power is limited, the speeds of grid search and Bayesian optimization are prolonged. If the computing power allows, it is recommended to use grid search or Bayesian optimization algorithms to select better hyperparameters to improve the accuracy.

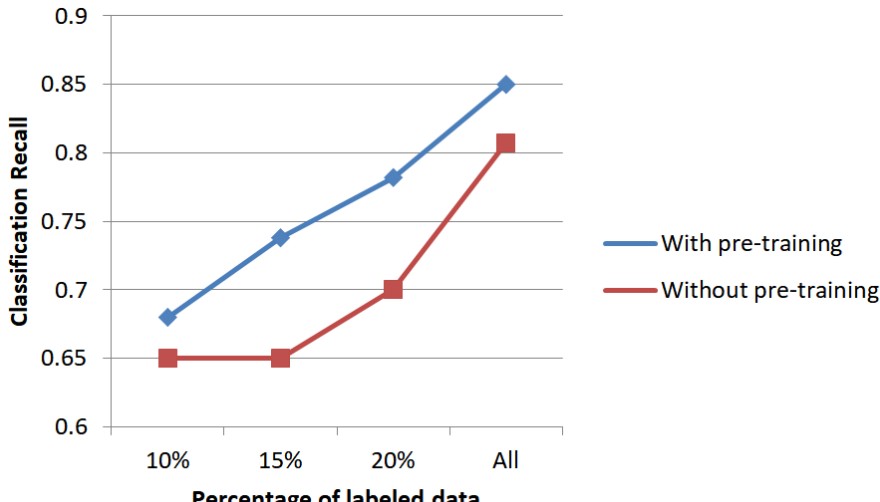

**Figure 9.** Visualization of the classification recall comparison with and without momentum contrastive learning.

## 4. Conclusions and Future Work

Wafer bin maps play an important role in yield improvement of semiconductor manufacturing. Automatic defect classification on WBMs can help engineers quickly locate problems on the production line and make timely adjustments. At present, most WBM automatic classification models require training with a large amount of labeled WBM data, and manual labeling is time-consuming. In practice, there are large-scale unlabeled WBMs that are not exploited. Therefore, determining how to use large-scale unlabeled data information to reduce the demand for labeled data is a key problem to be solved in semiconductor manufacturing.

We proposed a two-step WBM automatic classification framework consisting of self-supervised pretraining and fine-tuning stages. Firstly, we introduced a self-supervised pretraining method based on momentum contrastive learning. Positive sample pairs were generated through data augmentation, while negative sample pairs were created using a queue of samples. By optimizing the contrastive loss, the model learned to extract features that characterize WBMs. Through ablation experiments, we demonstrated that momentum contrastive learning can effectively learn feature representations from a large amount of unlabeled wafer maps, improving the model's classification performance and reducing the need for labeled data.

Additionally, to further enhance the training efficiency of labeled data, we proposed a fine-tuning method based on prototypical networks. The prototypical network is a mechanism for few-shot learning, where samples are classified based on their distances to the centers of each class. We compared our framework with two existing WBM defect classification works, and the results show that our method outperforms the others in terms of classification performance. Furthermore, we suggest removing labeled data with "none" defect type and only using eight specific defect patterns from the WM-811K dataset for fine-tuning. This approach improved the classification performance of WBMs with specific defects.

Future work will focus on researching self-supervised learning and classification methods for WBMs with multiple defects, as in most practical cases, a wafer usually has multiple defects. A single-defect classification model cannot accurately classify wafer maps with multiple defects. Therefore, a practical issue is designing a multidefect classification model and combining it with self-supervised learning to enhance the model's representation ability. In future research, we will explore a framework for self-supervised learning and multidefect classification of WBMs.

**Author Contributions:** Conceptualization, Z.H. and D.N.; methodology, Y.W., D.N. and Z.H.; software, Y.W.; validation, Y.W., D.N. and Z.H.; formal analysis, Y.W., D.N. and Z.H.; investigation, Y.W.; resources, Y.W.; data curation, Y.W.; writing—original draft preparation, Y.W. and D.N.; writing—review and editing, Z.H. and D.N.; visualization, Y.W., D.N. and Z.H.; supervision, D.N. and Z.H.; project administration, D.N.; funding acquisition, D.N. All authors have read and agreed to the published version of the manuscript.

**Funding:** This research was funded by National Science Foundation China grant No. 62173298.

**Institutional Review Board Statement:** Not applicable.

**Informed Consent Statement:** Not applicable.

**Data Availability Statement:** Not applicable.

**Acknowledgments:** The authors would like to thank the Intel Corporation for supporting the research.

**Conflicts of Interest:** The authors declare no conflict of interest.

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
