# Peer review of "A Momentum Contrastive Learning Framework for Low-Data Wafer Defect Classification in Semiconductor Manufacturing"

_applsci, doi:10.3390/app13105894_

Round 1
Reviewer 1 Report
This article includes a momentum contrastive learning approach for low-data wafer bin map classification. The paper is well organised and scientifically sound. The results are presented correctly and outperform similar approaches. In addition, all references are relevant. However, this work could be improved in a number of ways:
1) Further references are needed to support some of the statements in the Introduction section, e.g. "Manual labeling is time-consuming and can be subject to variations due to the subjective judgments of different engineers" or "Self-supervised learning is a type of machine learning where a model learns to represent data in a way that is useful for downstream tasks without requiring direct supervision or labeled data."
2) In section 2.2, the impulse coefficient (m) is set at m=0.999 and is said to work better than using smaller values. Could you justify this by testing other values?
3) In section 3.1, why do you use this distribution of data (60% for training and 40% for testing)? Why do you not perform any kind of cross-validation?
4) In section 3.1, how do you determine the hyperparameters of the model (batch size, learning rate, epochs,...)? The use of a hyperparameter tuning algorithm (grid search, bayesian,...) is highly recommended.
5) Please check figure 11. According to the description, the y-axis label is not correct.
Reviewer 2 Report
This paper has demonstrated the novel machine learning framework which could be utilizied for wafer defect classification, especially in semiconductor manufacturing. I believe this manuscript is well prepared and well organized, but a few minor revision is needed. Therefore, I would like to give this manuscript "accept with minor revision".
1) In Fig 11, the accuracy has been improved to some extent. However, in order to accomplish more higher accuracy, what can we do? Please kindly add this details in the part 3.3.
2) Please kindly add some reference for the following sentence. "ICCAD’21 proposed a shared backbone of self-supervised and few-shot learning and united the two learners in one period. "
3) Please revise '4. Conclusion' as '4. Conclusion and future work'.
Thanks.
I think the authors have already prepared good english writing.
